# EEG Oscillations in Specific Frequency Bands Are Differently Coupled with Angular Joint Angle Kinematics during Rhythmic Passive Elbow Movement

**DOI:** 10.3390/brainsci12050647

**Published:** 2022-05-14

**Authors:** Takako Suzuki, Makoto Suzuki, Kilchoon Cho, Naoki Iso, Takuhiro Okabe, Toyohiro Hamaguchi, Junichi Yamamoto, Naohiko Kanemura

**Affiliations:** 1School of Health Sciences, Saitama Prefectural University, 820 Sannomiya, Koshigaya 343-8540, Saitama, Japan; hamaguchi-toyohiro@spu.ac.jp (T.H.); kanemura-naohiko@spu.ac.jp (N.K.); 2Faculty of Health Sciences, Tokyo Kasei University, 2-15-1 Inariyama, Sayama 350-1398, Saitama, Japan; cho-k@tokyo-kasei.ac.jp (K.C.); iso-n@tokyo-kasei.ac.jp (N.I.); okabe-t@tokyo-kasei.ac.jp (T.O.); 3Department of Psychology, Keio University, 2-15-45 Mita, Minato-ku 108-8345, Tokyo, Japan; yamamotj@flet.keio.ac.jp

**Keywords:** electroencephalogram, oscillation, passive movement, event-related desynchronization, event-related synchronization

## Abstract

Rhythmic passive movements are often used during rehabilitation to improve physical functions. Previous studies have explored oscillatory activities in the sensorimotor cortex during active movements; however, the relationship between movement rhythms and oscillatory activities during passive movements has not been substantially tested. Therefore, we aimed to quantitatively identify changes in cortical oscillations during rhythmic passive movements. Twenty healthy young adults participated in our study. We placed electroencephalography electrodes over a nine-position grid; the center was oriented on the transcranial magnetic stimulation hotspot of the biceps brachii muscle. Passive movements included elbow flexion and extension; the participants were instructed to perform rhythmic elbow flexion and extension in response to the blinking of 0.67 Hz light-emitting diode lamps. The coherence between high-beta and low-gamma oscillations near the hotspot of the biceps brachii muscle and passive movement rhythms was higher than that between alpha oscillation and passive movement rhythm. These results imply that alpha, beta, and gamma oscillations of the primary motor cortex are differently related to passive movement rhythm.

## 1. Introduction

Passive movements are often used during rehabilitation to improve the range of motion, motor function, and proprioceptive function [1,2,3]. According to topographic studies, the primary motor cortex, supplementary motor area, and premotor cortex are activated during active movements [4]. Additionally, previous studies have investigated the time course of changes in cortical oscillations during active movements [5]. During movement execution by humans, the power of alpha and beta oscillations decreases in the contralateral sensorimotor cortex 2 s before movement [6,7,8,9]. These phenomena are called event-related desynchronization. Furthermore, the increase in gamma oscillation power is accompanied by alpha desynchronization and beta desynchronization in the contralateral sensorimotor area during active movements [5,10], which is referred to as event-related synchronization. At the end of the movement, recovery of alpha and beta oscillations occurs (alpha synchronization and beta synchronization) [7,8]. In addition to gamma synchronization, alpha desynchronization and beta desynchronization have been related to motor preparation. In contrast, post-movement alpha synchronization and beta synchronization have been related to sensory afferent activities because they disappeared with ischemic nerve block [5,11] and the process of returning the motor cortex to a resting state [8].

Despite the absence of voluntary planning and preparation during passive movement, the sensorimotor cortex, primary motor area, supplementary motor area, and primary and secondary somatosensory areas are activated during passive movements, resembling active movement [12,13]. The primary motor cortex neurons receive and process sensory afferent inputs from muscle spindles without generating any active movement and are activated during passive movements [14,15,16,17]. Additionally, despite the absence of voluntary movement during passive movement, afferent somatosensory components (skin mechanoreceptors, muscle spindles, and joint receptors) produce premovement alpha desynchronization and beta desynchronization [18,19].

Previous studies of cortical oscillations in accordance with active and passive movements implied that the sensorimotor cortex is activated by both active and passive movements [12,13], cortical oscillations changed during active movements [5], and post-movement alpha synchronization and beta synchronization occurred during passive movements [5,11]. However, despite being responsible for the motor command and receiving the sensory afferent inputs, the time courses of alpha, beta, and gamma oscillations with passive movement rhythm remain controversial. The relationship between passive movement rhythm and alpha, beta, and gamma oscillations in the sensorimotor cortex is still unclear. Knowledge of brain oscillations in accordance with passive movements may contribute to an evidence-based approach to passive movement training in rehabilitation settings. We predicted that if the primary motor cortex receives sensory input from the somatosensory cortex during passive movement, then the cortical oscillations should be related to the passive movement rhythm. Therefore, we hypothesized that the alpha, beta, and gamma oscillations in the primary motor cortex would change differently in accordance with the passive movement rhythm. Therefore, we aimed to quantitatively identify the relationship between brain oscillations and passive movement rhythm.

## 2. Materials and Methods

### 2.1. Participants

Our target sample size was based on 95% statistical power to detect changes in the elbow angle and brain oscillation with an effect size of 0.70 and a two-sided α-level of 0.05. The input of the aforementioned parameters into the Hulley matrix [20] yielded a sample size of 20. Accordingly, we recruited 20 healthy, neurologically intact participants (6 men and 14 women; age range: 20–47 years; mean age ± standard deviation (SD): 27.1 ± 9.1 years) to undergo behavioral and electroencephalography (EEG) measurements. The screening revealed that none of the patients were at risk for adverse events caused by transcranial magnetic stimulation [21]. Additionally, none of the participants were prescribed medications or diagnosed with psychiatric or neurological diseases. We confirmed right-hand dominance with the Edinburgh Handedness Inventory [22] and recorded a mean laterality quotient score of 0.9 points (SD: 0.1 points). Our experimental procedures were approved by the Research Ethics Committee of the Tokyo Kasei University and the Saitama Prefectural University and followed the principles of the Declaration of Helsinki. All participants provided written informed consent prior to their participation.

### 2.2. Hotspot Detection

Participants were seated with the elbow flexed at 90° and the right forearm strapped to an armrest. We recorded motor-evoked potentials (MEPs) in the right biceps brachii muscle using double differential surface electrodes (FAD-DEMG1; 4 Assist, Tokyo, Japan). Prior to recording the MEPs, we cleaned the skin overlying the biceps brachii muscle with alcohol to reduce its electrical resistance. MEP signals were amplified ×100, bandpass-filtered at 10 to 2000 Hz, digitized at 10 kHz with a PowerLab system (ADInstruments, Dunedin, New Zealand), and stored in magnetic media.

Monophasic transcranial magnetic stimulation was delivered to the scalp surface through a figure-of-eight coil (internal diameter of each wing: 70 mm) using the Magstim 200^2^ (Magstim, Dyfed, UK) stimulator. After placing a tight-fitting cap over the head of the participant, we drew the intersecting nasion–inion and interaural lines on the cap with a marker pencil to localize the vertex (Cz) in accordance with the 10–20 International System. To induce current flow in the left brain from the posterior–lateral to the anterior–medial direction, we placed the coil tangentially to the scalp while holding the handle pointing backward and sideways approximately 45º from the midline. First, we measured the isometric maximum voluntary contraction such that the participants were instructed to flex the elbow as much as possible for 5 s. Then, we visually detected the optimal coil position to elicit maximum MEPs in the right biceps brachii muscle (“hotspot”) during 10% maximum voluntary contraction of the biceps brachii muscle and marked the location with a soft-tip pen.

### 2.3. Electroencephalography

After hotspot detection, we recorded EEG data before and during rhythmic passive movements using Polymate V (Miyuki Giken, Tokyo, Japan) with a gold-coated active electrode. After skin preparation, the EEG electrodes were placed over a nine-position grid (5 × 5 cm^2^); the center was oriented on the hotspot of the biceps brachii muscle (Figure 1).

EEG data were sampled at 1000 Hz and filtered from 0.15 to 200 Hz; we maintained electrode impedance ≤10 kΩ. EEG signals were referenced to the averaged recordings from the electrodes on the bilateral earlobes. Before rhythmic passive movements, we recorded a resting period of 35 s. Then, EEG data were recorded during rhythmic passive movements.

### 2.4. Passive Movement

We recorded movements at the elbow using an electric goniometer (FA-DL-263; 4Assist, Tokyo, Japan) mounted on the elbow joint. A positive angle value corresponded with elbow flexion. Signals from the linear potentiometer were digitized at 1000 Hz (Miyuki Giken, Tokyo, Japan). The wrist and finger were fixed to a plastic support board. Two light-emitting diode (LED) lamps—each blinking at 0.67 Hz for 100 ms—were placed 0.25 m and 0.75 m from the right acromion of the participant (Figure 1). Passive movements comprised 30 blocks of eight movement cycles, including elbow flexion and extension, with a 5 s to 7 s random break. A blue dot was constantly displayed on the screen for gaze fixation during the resting and passive movement period; it was located approximately 100 cm ahead of the participant. We instructed the participants to relax and look at the blue dot, to not count during the movement, and to keep their arms relaxed throughout the experiment. Before the start of passive movement trials, the assistants took the participant’s elbow and wrist and guided them in the appropriate direction several times in accordance with the blinking of the LED lamps to familiarize them with rhythmic passive movements. The assistants were instructed to perform rhythmic elbow flexions and extensions as directly and accurately as possible so that the index fingertip was near (0.25 m from the participant’s acromion) the LED and far (0.75 m from the participant’s acromion) the LED in response to the blinking of the LED lamps.

### 2.5. Data Analysis

We predicted that alpha, beta, and gamma oscillations would be differently related to the passive movement rhythm because the primary motor cortex receives sensory input from the somatosensory cortex and each oscillation band has a different role in sensory processing. Based on this prediction, the time–frequency analysis and coherence calculations were performed to measure the phase synchronization of the EEG power spectra to the time-locking passive movements. In the time–frequency analysis, the onset of movement was defined as the time point when the elbow angle exceeded the average of the baseline angle by 2 standard deviations (SDs) of the baseline. The epochs of 12,000 ms after movement onset were extracted from each block’s EEG and elbow angle data. Time–frequency analysis was performed using Morlet wavelet transforms [23] in frequencies between 8 and 80 Hz for each epoch. Additionally, EEG signal periods exceeding amplitudes of ±200 μV were detected to exclude the data, including eye blinking or muscle movement artifacts (99.3% of the raw data points were kept for further analysis). After artifact rejection, event-related power spectra were averaged across 30 epochs. Moreover, frequency bands of interest were selected in the alpha (8–12 Hz), low-beta (12–24 Hz), high-beta (24–30 Hz), low-gamma (30–60 Hz), and high-gamma (60–80 Hz) ranges. Then, the power spectra were evaluated for the outliers using Tukey’s fences [24]. The data with values more than 1.5 times the interquartile range were excluded from the datasets. After removing the outliers, the blank cells were linearly interpolated. Next, the power spectra and elbow angle for each participant were normalized using linear transformation to analyze a coherence between individual variability of the EEG power spectrum and the elbow angle. The data were expressed as a *Z score*: $Z\;score=\frac{\left(x_{i}\;–\;M\right)}{s}$, where $x_{i}$ refers to the sample EEG power spectra or elbow angle, M refers to the mean EEG power spectra or elbow angle, and s refers to the SD of the data. Moreover, the *Z scores* for EEG power spectrums were averaged from nine EEG channels to compensate for the low spatial accuracy of 9 electrodes for the hotspot of biceps brachii muscle as a region of interest. Furthermore, the *Z score* for EEG power spectrum and the elbow angle were low-pass-filtered at 1 Hz to homogenize the difference in temporal and spectral smoothing between the alpha, low-beta, high-beta, low-gamma, and high-gamma oscillations by reference to a previous study [5]. Then, coherence calculation based on the Fourier transform of 5-s epochs (0.2 Hz resolution) which was updated every 3.75 s (75% data overlapped) was computed between EEG oscillation and passive movement rhythm by reference to the study by Bourguignon et al. [25].

To conduct a comparison of differences in the coherence values of EEG oscillations and passive movement rhythm, the one-way repeated measures analysis of variance was performed. Furthermore, a post hoc analysis with the Bonferroni correction was performed to compare differences in the coherence between the alpha, low-beta, high-beta, low-gamma, and high-gamma oscillations. Data analysis was performed using EMSE (Miyuki Giken, Tokyo, Japan), the scipy package in the Python environment, and R 3.5.2 software (R Foundation for Statistical Computing, Vienna, Austria). Data are expressed as mean ± standard error of the mean (SEM). We defined statistical significance as *p* < 0.05.

## 3. Results

### 3.1. Consistency in Passive Movement Rhythms

The difference between passive elbow movement and the LED blinking cycles was −12.1 ± 3.1 ms, indicating a movement delay of 12.1 ms for LED blinking. The rhythmic passive movement procedure was successful because of the small SEM (i.e., 3.1 ms).

### 3.2. Time Course of Changes in the EEG Power Spectrum and Elbow Angle

Figure 2 depicts the time courses of changes in the Z scores of EEG oscillations to the time locking of the elbow angles during a 12 s movement block of eight movement cycles. The time courses of changes occurred in the mean ± SEM of the alpha-, beta-, and gamma-band oscillations and in the mean ± SEM of the elbow angles (Figure 2A–F) during the passive movement cycle. The Gray zone in Figure 2F denotes SEM of the elbow angle, but it is hardly visible because deviations in the range and rhythm of the elbow angle were extremely small. Seemingly, we observed that EEG oscillations, except for high-gamma oscillation, similarly oscillated to the passive movement rhythm with the same time-locked time scale.

### 3.3. Coherence between EEG Oscillations and Passive Movement Rhythm

The coherence between alpha-band oscillation and passive movement rhythm, low-beta-band oscillation and passive movement rhythm, high-beta-band oscillation and passive movement rhythm, low-gamma-band oscillation and passive movement rhythm, and high-gamma oscillations and passive movement rhythm were 0.164 ± 0.002, 0.167 ± 0.002, 0.171 ± 0.002, 0.172 ± 0.002, and 0.169 ± 0.002, respectively (Figure 3). The one-way repeated measures analysis of variance demonstrated significant differences in the coherence among alpha, low-beta, high-beta, low-gamma, and high-gamma oscillations (F = 4.102, *p* = 0.0025). During post hoc testing, using the Bonferroni correction, the coherence value of alpha oscillation was significantly lower than that of the high-beta and low-gamma oscillations (alpha vs. low beta, *p* = 1.000; alpha vs. high beta, *p* = 0.013; alpha vs. low gamma, *p* = 0.005; alpha vs. high gamma, *p* = 0.349; low beta vs. high beta, *p* = 0.519; low beta vs. low gamma, *p* = 0.267; low beta vs. high gamma, *p* = 0.519; high beta vs. low gamma, *p* = 1.000; high beta vs. high gamma, *p* = 1.000; low gamma vs. high gamma, *p* = 1.000) (Figure 3).

## 4. Discussion

We measured changes in cortical oscillations during rhythmic passive movements to test the hypothesis that alpha, beta, and gamma oscillations of the primary motor cortex should be differently related to the passive movement rhythm. Our findings suggested the following: the coherence between high-beta and low-gamma oscillations near the hotspot of the biceps brachii muscle and passive movement rhythms was higher than that between alpha oscillation and passive movement rhythm. These results imply that each cortical oscillation near the hotspot of the biceps brachii muscle differently changed in accordance with the passive movement rhythm.

Previous studies [11,26,27] noted that alpha-band and low-beta-band oscillations (15–25 Hz) increased after passive movements. These increments in oscillations required input from the somatosensory afferents because this phenomenon was eliminated by an ischemic nerve block [11]. Additionally, previous studies demonstrated an association between two peaks of the magnetoencephalography response and passive finger movements from 30 to 100 ms after movement onset [12]. The earliest component was estimated to be in the primary motor cortex. In contrast, the second component was estimated to be in the primary motor cortex as well as in the supplementary motor area, posterior parietal cortex over the hemisphere contralateral to the movement, and secondary somatosensory cortex of both hemispheres [12]. Our study found that alpha, beta, and gamma oscillations were differently related to passive movement rhythm. This is the first new observation in our study. During rhythmic passive movements, the sensorimotor cortex neurons receive sensory afferent inputs from skin mechanoreceptors, muscle spindles, and joint receptors [14,15,16,17]. During our study, the sensory inputs continuously reached the sensorimotor cortex because of the rhythmic passive movements. One possible explanation for cortical oscillations during rhythmic passive movements is that changes in alpha, beta, and gamma oscillations may be evoked by sensory afferents.

Additionally, previous studies regarding rhythmic stimulation without actual movements reported an association between the time course of beta oscillation in the sensorimotor cortex and the predictive timing of upcoming external rhythmic visual [28] and auditory [29] stimuli. Beta synchrony in the sensorimotor cortex increased from the start of the informative external cue and diminished afterward [28]; therefore, beta oscillation may represent processes related to the prediction and integration of sensory information because of the presence of sensory input of the fingers during movement [5]. Our study also found that the coherence between high-beta and low-gamma oscillations and passive movement rhythm were relatively high. This is the second new observation in our study. One possible explanation for high-beta oscillation during rhythmic passive movement is that high-beta oscillation may represent processes related to the prediction of the next passive movement cycle based on sensory information.

Previous studies also reported a close relationship between gamma amplitudes and movement sequences [30]. Additionally, previous studies noted that beta-band oscillations were associated with inhibitory γ-aminobutyric acid (GABAergic) interneurons in several cortical regions, including the primary motor area [31,32], and the increase in gamma oscillation power was accompanied by beta desynchronization [5,10]. These results imply that beta and gamma oscillations have opposing roles in the sensorimotor cortex. Beta oscillation inhibits, whereas gamma oscillation facilitates, motor preparation and execution [5]. According to previous studies of active movements, gamma amplitudes were conversely modulated to the beta amplitude during rhythmic active finger tapping [5] and walking [33,34]. During our study, the high-beta and low-gamma bands similarly oscillated to passive movement (i.e., coherences between high-beta and low-gamma oscillations and passive movement were relatively high). Therefore, both high-beta and low-gamma oscillations may be related to passive movement rhythm. Further research is necessary to investigate the reciprocal functions of beta and gamma oscillations with rhythmic active and passive movements.

The time course of alpha oscillation previously resembled that of beta oscillation in sensorimotor areas [5]. However, the coherence between alpha oscillation and passive movement was lower than that between high-beta and low-gamma oscillations and passive movement rhythm in our study. This necessitates further study to investigate the detailed function of alpha oscillation during rhythmic passive movements.

Several limitations of our study need to be considered. There were only nine EEG electrodes with the center at the hotspot of the biceps brachii muscle in our study. Therefore, we did not investigate specific cortical networks in the sensorimotor cortex. Further studies are necessary to investigate the specific cortical networks based on whole-brain EEG electrodes during passive movements to clarify the functions of alpha, beta, and gamma oscillations. Additionally, muscle activity during passive movement was not directly confirmed during our study; however, the EEG signal more than ±200 μV that included eye blinks or muscle movement artifacts was excluded. Further studies using both EEG and electromyography recordings during passive movements are required to understand the oscillatory changes in the time course of the passive movement cycle and to elucidate the mechanisms underlying the relationship between EEG oscillations and passive movement rhythm.

## 5. Conclusions

Rhythmic passive movements were differently associated with alpha, beta, and gamma oscillations at the primary motor cortex. Our findings imply that passive movements rely on the nonequivalent activation of cortical neurons in the primary motor area.

## Figures and Tables

**Figure 1 brainsci-12-00647-f001:**
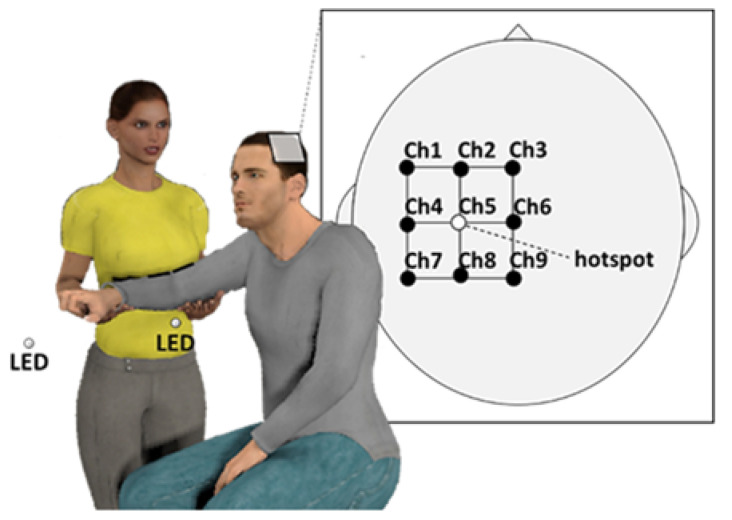
Experimental design used to evaluate rhythmic passive movements. EEG electrodes were placed over a nine-position grid (5 × 5 cm^2^); the center was oriented on the TMS hotspot of the biceps brachii muscle. Two LED lamps were placed 0.25 m and 0.75 m from the right acromion of the participant. Passive movements included elbow flexion and extension in response to the blinking of the LED lamps: EEG, electroencephalography; TMS, transcranial magnetic stimulation; LED, light-emitting diode. Ch, channel.

**Figure 2 brainsci-12-00647-f002:**
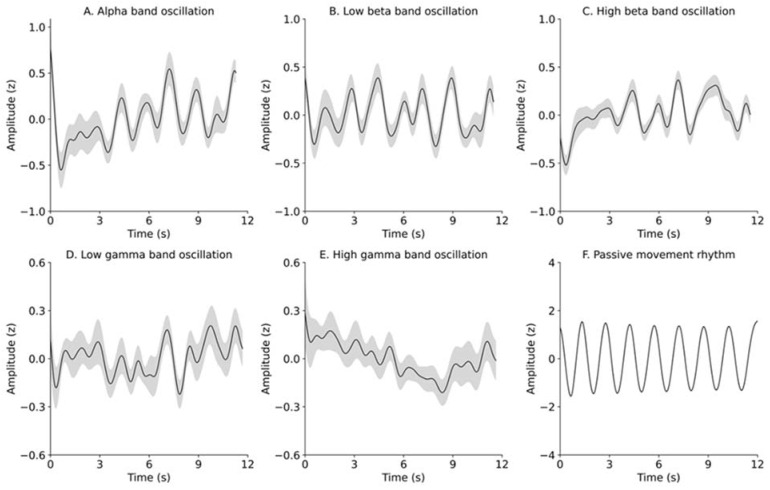
Time courses of changes in the mean ± SEM of alpha (**A**), low-beta (**B**), high-beta (**C**), low-gamma (**D**), or high-gamma (**E**) oscillations, and in the mean ± SEM of the elbow angle (**F**). Black lines and gray zones denote the mean and SEM of alpha (**A**), low-beta (**B**), high-beta (**C**), low-gamma (**D**), or high-gamma (**E**) oscillations. Black dashed lines and gray zones denote the mean and SEM, respectively.

**Figure 3 brainsci-12-00647-f003:**
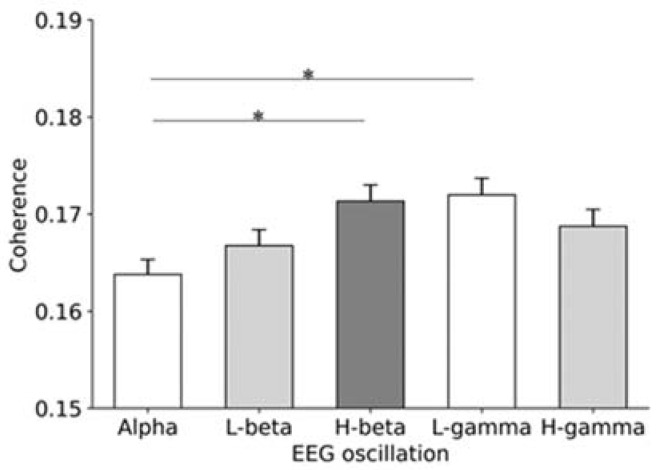
The mean ± SEM of the coherence values in alpha, low-beta, high-beta, low-gamma, and high-gamma oscillations with passive movement rhythm. The coherence value of the alpha band was significantly lower than those of the high-beta and low-gamma bands (alpha vs. high beta, *p* = 0.013; alpha vs. low gamma, *p* = 0.005). SEM, standard error of the mean: L-beta, low beta; H-beta, high beta; L-gamma, low gamma; H-gamma, high gamma. *: *p* < 0.05.

## Data Availability

Raw data were generated at Tokyo Kasei University. Derived data supporting the findings of this study are available from the corresponding author, T.S. or M.S.

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
