# Peer review of "EEG Oscillations in Specific Frequency Bands Are Differently Coupled with Angular Joint Angle Kinematics during Rhythmic Passive Elbow Movement"

_brainsci, 2022, doi:10.3390/brainsci12050647_

Round 1

Reviewer 1 Report

This study analyzed changes in cortical oscillations during rhythmic passive movements and their relations with passive movement rhythm. It suggested meaningful results for passive movement but need further revision.

Reviewer 2 Report

Manuscript brainsci-1652760 aims to investigate the relationship between brain oscillations in the alpha, beta and gamma bands and the movement rhythm during passive elbow movement.

Even if I sincerely feel that the presented study has the potential for an interesting paper, manuscript brainsci-1652760 suffers to my opinion from major flaws that regretfully make the rationale not entirely clear to the reader, the methods somewhat confusing on many aspects, and the results not accurate enough for meaningful interpretation and discussion.

Below are my main comments and questions to the authors organized by manuscript section that hope will help to improve the manuscript:

- Throughout the manuscript, it is unclear why the authors mean by "phase" of the passive movements. Beyond a simple semantic issue, the lack of a precise definition is at the expense of a clear understanding of both the methods and some results.

- Introduction, lines 50-51: To my opinion, this statement is not fully supported by the literature. This needs clarification.

- Introduction, lines 57-58: What does the authors mean by "the beta synchronization is highest and lowest at the beginning of a movement cycle"? It appears very general and even contradictory. Please rephrase.

- Introduction, lines 71-78: The authors must provide an elaborated rationale to justify in what the interrelationship between alpha, beta, and gamma oscillation according with the phase of rhythmic passive movements remains controversial. In the actual form of manuscript brainsci-1652760, the justification of the study is not well demonstrated.

- Introduction, lines 79-88: The hypotheses are very general and not well defined. This section needs to be significantly improved.

- Methods, 124-125: I am not expert with TMS but, even if I clearly understand the interest of identifying a hotspot for subsequent EEG positioning and analysis, I do not fully understand the procedure and to which extent the optimal coil position to elicit maximum MEPs in the right FDI muscle can be taken as the hotspot of the biceps brachii muscle.

- Methods, line 131: EEG is recorded with 9-electrode set. This raises major concerns regarding the reliability of EEG measurements. This needs to be addressed at least as a major limitation of the study.

- Methods, lines 154-155: What was the purpose of the electrodes placed  above and below the left eye? More generally, how many movements were rejected for each participants before analysis?

- Methods, lines 180-181: I agree with the authors that the proposed additional - but separate - EMG analysis provides experimental evidence that the level EMG activation can be reasonably considered as negligible in similar conditions to those during which EEG and movement kinematics were recorded, but the EMG results should be considered and contextualized with more caution especially in the methods section lines 270-271 and in the results section lines 477-478.

- Methods, lines 200-201: Please improved the "way to define and describe Morlet wavelets for time-frequency analysis" (Cohen, 2019; https://doi.org/10.1016/j.neuroimage.2019.05.048).

- Methods, lines 202-203: What do the authors mean by "were analyzed and logarithmically transformed". The nature of the processing needs to be more precisely specified.

- Methods, lines 204-212: I do not understand in what Eq. 1 corresponds to the calculation of the ERD and ERS "of the resting period EEG". Moreover, from what is written it is  not clear that the authors properly calculated the ERD/ERS as recommended by Pfurtscheller (1992; https://doi.org/10.1016/0013-4694(92)90133-3). Again, there is a lack a clarity that needs to be addressed.

- Methods, lines 219-241: I regret but, despite multiple readings of manuscript brainsci-1652760, this part of the methods section is very unclear to me. Some definitions are missing and there is a lack of explanations on the meaning of the obtained metrics with regards the objectives of the study. In the absence of such clear information, the related results are not meaningful and are conversely very difficult to interpret. This is reinforced by the fact that the analysis relies on a very important number of very diverse dependent variables. I think that the strength of the study would be greatly improved if the authors focused on a more limited number of well-justified variables fully reliable regarding the purpose to the study.

- Results: Significant improvement in the description of the results is required. In its actual form, it is very confusing, remains somewhat descriptive and is not always supported by the data both in the main text, in the figures and in the figures' legends. Hence the opening section of the discussion is not sufficiently supported by the findings.

- Discussion & Conclusion: To my opinion, the discussion must be rewritten, to be closely related to the subject matter and critically focused upon clearer results. As the discussion is currently presented, the conclusion is not fully supported by the data, which make it too speculative and not yet totally convincing.

Round 2

Reviewer 1 Report

The authors have answered all of my questions and revised the manuscript accordingly. I have no more comments.

Reviewer 2 Report

The authors have made an important effort to address the major criticisms raised by Reviewer 2 and myself. Thanks to the improvements that have been made to manuscript brainsci-1652760, I acknowledge that the aim of the study is clearer in the revised version. However, I am afraid that I) the proposed methodology is not fully suitable to achieving the stated objective of the study and ii) the analysis suffers to my opinion from a major flaw that questions the validity and the interpretability of the results.

Firstly, I understand from the rationale provided in the revised version that the primary aim of the study is to investigate to what extent the modulation of the reactivity of brain rhythms (in alpha, beta and gamma frequency bands) is linked to that of angular kinematics during passive elbow flexion-extension movements. In view of previous studies between on the coupling between cortical activity and the kinematics of movements, I think that the appropriate method is not the one used the study but rather in corticokinematic coherence (e.g., Bourguignon et al. (2015; https://dx.doi.org/10.1016%2Fj.neuroimage.2014.11.026)). Without using this approach, the analysis remains very descriptive and has major flaws that make the results only slightly meaningful.

Secondly, the information provided by the authors in the revised version concerning wavelet analysis parameterization confirms that both the temporal smoothing and the spectral smoothing were different between the alpha, beta and gamma frequency bands. This makes the results - in particular the observed time differences between EEG oscillations - not comparable, and seriously calls into question the interpretation and relevance of the results.

Finally, due to the low spatial accuracy inherent in EEG recordings, I am not convinced that the analysis of individual power spectrum and of the so-called "CoG" is reliable, even more by using an EEG system equipped with only nine electrodes. I strongly suggest the authors to perform a ROI-based analysis in each targeted frequency band.

Thus, despite the improvements made to the revision version of manuscript brainsci-1652760, I think that the proposed analysis is highly questionable, and I am still not convinced that the conclusions drawn by the authors are adequately supported by the data.

Round 3

Reviewer 2 Report

I would like to congratulate the authors for the improvements they have made to the manuscript. I think that some of the Discussion points remain somewhat speculative, but the conclusions can now be considered sufficiently supported by the results. In my opinion, this revised version of the article is incomparably better than the original version.

I suggest the authors to change the title to: "EEG oscillations in specific frequency bands are differently coupled with angular joint angle kinematics during elbow rhythmic passive movement".

In section 3.3, I invite the authors to add a figure of typical recordings with elbow angle on one panel and the time-frequency map of the EEG oscillations on another panel.

In the current Figure 2 :

  • The grey area that is supposed to represent the SEM of elbow is missing.
  • Why is the time scale of the angle joint to the same to that of EEG oscillations? This needs to be corrected.
